# Product of Hyperbolic Spaces
# for Vision–Language Representation Learning

## Abstract

Vision–language representation learning bridges the gap between visual and linguistic modalities by capturing their shared semantics. Hyperbolic spaces capture tree-like hierarchy within a concept family, whereas they lack canonical operations to represent the compositionality across different concept families. To resolve this dilemma, we propose PHyCLIP, which employs an $\ell_1$-*Product* metric on a Cartesian product of *Hy*perbolic factors. Our key insight is that the cross-family compositionality can be naturally captured by an $\ell_1$-product metric, analogous to a Boolean algebra, while the intra-family hierarchy is embedded into each hyperbolic factor. Experiments on zero-shot classification, retrieval tasks, and hierarchical classification demonstrate that PHyCLIP outperforms existing single-space approaches and offers more interpretable representations of compositionality in visual scenes and linguistic descriptions.

## 1 Introduction

Vision–language models such as CLIP (Radford et al., 2021) have provided representations that can be transferred across tasks. However, mapping each image or text to a single-vector embedding makes it difficult to simultaneously capture two fundamental semantic structures: *hierarchy* (*is-a* relations within a family) and *compositionality* (conjunctive co-occurrence across families). Linguistic concepts linked by *is-a* (entailment) relations form a hierarchical structure (e.g., "chihuahua" $\preceq$ "dog" $\preceq$ "mammal"), often treated as a tree (Miller, 1995). Conventional Euclidean embeddings struggle to represent such structure due to the exponential growth of nodes in a tree, and hyperbolic space effectively captures hierarchical relations (Nickel & Kiela, 2017; Ganea et al., 2018; Desai et al., 2023; Pal et al., 2025). At the same time, a text or image often mentions a composition of multiple concepts from different families, which can be viewed as a conjunction (e.g., "a dog in a car" $\simeq$ "dog" $\wedge$ "car") and treated as another type of entailment (e.g., "a dog in a car" $\preceq$ "dog", "car"). Some prior works suggest that representations are disentangled if a transformation group acting on the space is decomposed into a direct product of subgroups (Higgins et al., 2018); such group can be easily found on Euclidean space, but not on hyperbolic space. In short, the suitability of hyperbolic embeddings for representing cross-family compositionality remains unclear.

To resolve this dilemma, we introduce PHyCLIP, which leverages an $\ell_1$-product metric on a Cartesian product of hyperbolic factors. Each hyperbolic factor encodes a tree-like intra-family hierarchy, while the $\ell_1$ aggregation captures Boolean-like cross-family compositionality, unifying two semantic structures in a single framework. Unlike prior models with mixed-curvature products (Gu et al., 2019; Wang et al., 2024; Gao et al., 2025), our design employs an $\ell_1$-product metric rather than a Riemannian ($\ell_2$) product metric and restricts curvature of each factor to remain negative. Experiments on zero-shot classification, retrieval, and hierarchical evaluation benchmarks demonstrate consistent gains, particularly on fine-grained datasets where compositional reasoning is essential.

Submitted to 39th Conference on Neural Information Processing Systems (NeurIPS 2025). Do not distribute.

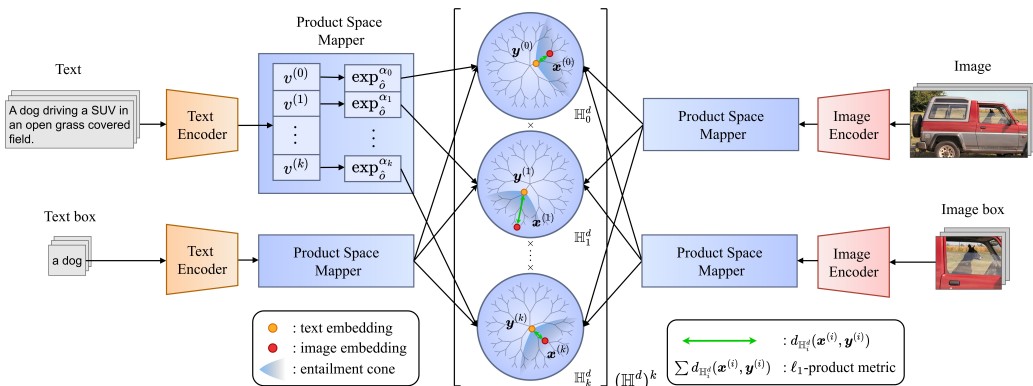

Figure 1: **Overview of PHyCLIP.** Images and texts are encoded as points $\boldsymbol{X}$ in an $\ell_1$-product metric space of hyperbolic factors, $(\mathbb{H}^d)^k$, that is, as tuples of points $\boldsymbol{x}^{(i)}$ in hyperbolic spaces $\mathbb{H}_i^d$, where their distance is defined by the sum of hyperbolic distances. The entailment relations $\boldsymbol{X} \preceq \boldsymbol{Y}$ are encoded using entailment cones as $\boldsymbol{x}^{(i)} \in C(\boldsymbol{y}^{(i)})$ within hyperbolic factors $\mathbb{H}_i^d$.

## 2 Method

### 2.1 Motivation and Architecture.

Images and texts often describe a composition of multiple concepts from different families (e.g., "a dog in a car"). Such composition has been represented by the conjunction (i.e., *AND* operation) in Boolean algebra, vector addition in Euclidean embeddings (Mikolov et al., 2013), or the product of probabilistic models (Vilnis & Mccallum, 2015). However, hyperbolic space lacks such canonical operations since Möbius addition in hyperbolic spaces (Ungar, 2008) is not aligned with standard vector addition or Boolean structures (Higgins et al., 2018). Motivated by this, we consider an $\ell_1$-product metric on a Cartesian product of multiple hyperbolic factors, referred to as an $\ell_1$-product metric space $(\mathbb{H}^d)^k$. This space is designed to simultaneously capture tree-like intra-family hierarchies using hyperbolic factors and Boolean-like cross-family compositionality using the $\ell_1$-product metric (see Appendix A for theoretical foundations). Figure 1 illustrates the overall framework. Given an image $I$ and text $T$, we extract a $kd$-dimensional feature vector, split it into $k$ segments, and lift each segment into a hyperbolic factor $\mathbb{H}_i^d$, yielding the embedding $\boldsymbol{X} = (\boldsymbol{x}^{(1)}, \ldots, \boldsymbol{x}^{(k)}) \in (\mathbb{H}^d)^k$.

We consider that an image entails a text ($I \preceq T$). Following HyCoCLIP (Pal et al., 2025), we also leverage box-level supervision: image crops $I^{\text{box}}$ and text spans $T^{\text{box}}$ induce relations $I \preceq I^{\text{box}}$, $T \preceq T^{\text{box}}$, and $I^{\text{box}} \preceq T^{\text{box}}$, providing additional hierarchical constraints. We denote the embeddings of images $I$ and texts $T$ as $\boldsymbol{I}$ and $\boldsymbol{T}$, respectively. Let $B$ denote the index set of instances in a mini-batch; we write the mini-batch of images as $\{\boldsymbol{I}_b\} = \{\boldsymbol{I}_b\}_{b \in B}$ for brevity.

To represent each hyperbolic factor $\mathbb{H}_i^d$, we adopt the Lorentz model with a learnable curvature $\alpha_i$ (Cannon et al., 1997; Nickel & Kiela, 2018; Lee, 2018). We introduce the distance on the $\ell_1$-product metric space $(\mathbb{H}^d)^k$ and the normalized version as

$$d_{(\mathbb{H}^d)^k}(\boldsymbol{X}, \boldsymbol{Y}) = \sum_{i=1}^k d_{\mathbb{H}_i^d}(\boldsymbol{x}^{(i)}, \boldsymbol{y}^{(i)}), \quad d_{\text{avg}}(\boldsymbol{X}, \boldsymbol{Y}) = \tfrac{1}{k} d_{(\mathbb{H}^d)^k}(\boldsymbol{X}, \boldsymbol{Y}). \tag{1}$$

### 2.2 Training Objectives.

To encourage an embedding $\boldsymbol{X}_b$ to attract the pair $\boldsymbol{Y}_b$, while repelling others $\boldsymbol{Y}_a$ for $a \neq b$, we use a standard InfoNCE loss (Radford et al., 2021; Desai et al., 2023; Pal et al., 2025):

$$L_{\text{cont}}(\{\boldsymbol{X}_b\}, \{\boldsymbol{Y}_b\}) = -\sum_{b \in B} \log \frac{\exp(-d_{\text{avg}}(\boldsymbol{X}_b, \boldsymbol{Y}_b)/\tau)}{\sum_{a \in B} \exp(-d_{\text{avg}}(\boldsymbol{X}_b, \boldsymbol{Y}_a)/\tau)} \tag{2}$$

where $\tau$ is a learnable temperature parameter. We average this loss over known pairs and get $\mathcal{L}_{\text{cont}}$.

$$\mathcal{L}_{\text{cont}} = \tfrac{1}{4}(L_{\text{cont}}(\{\boldsymbol{I}_b\}, \{\boldsymbol{T}_b\}) + L_{\text{cont}}(\{\boldsymbol{T}_b\}, \{\boldsymbol{I}_b\}) + L_{\text{cont}}(\{\boldsymbol{I}_b^{\text{box}}\}, \{\boldsymbol{T}_b^{\text{box}}\}) + L_{\text{cont}}(\{\boldsymbol{T}_b^{\text{box}}\}, \{\boldsymbol{I}_b^{\text{box}}\})) \tag{3}$$

We also employ hyperbolic entailment cones to capture the entailment relationships (Ganea et al., 2018). In hyperbolic space $\mathbb{H}^d$, each point $\boldsymbol{y}$ defines a cone $C(\boldsymbol{y})$ with half-aperture $\omega(\boldsymbol{y})$, to which

## Table 1: Zero-shot image classification evaluation.

| | w/ boxes | ImageNet | CIFAR-10 | CIFAR-100 | SUN397 | Caltech-101 | STL-10 | Food-101 | CUB | Cars | Aircraft | Pets | Flowers | DTD | EuroSAT | RESISC45 | Country211 |
|---|---|---|---|---|---|---|---|---|---|---|---|---|---|---|---|---|---|
| | | General datasets | | | | | | Fine-grained datasets | | | | | | Specialized datasets | | | |
| **ViT B/16** CLIP | | 39.36 | 75.09 | 48.57 | 51.48 | 73.63 | 92.54 | 50.59 | 13.40 | 7.66 | 2.42 | 46.44 | 19.00 | 23.19 | 35.26 | 42.60 | 5.20 |
| MERU | | 37.49 | 75.61 | 46.80 | 49.54 | 71.19 | 93.38 | 52.88 | 10.52 | 7.49 | 3.05 | 44.11 | **22.94** | 21.70 | **39.52** | 41.09 | 4.74 |
| HyCoCLIP | ✓ | 42.93 | 88.51 | 57.68 | 54.23 | 75.55 | 94.55 | 51.72 | 12.86 | 9.98 | **4.41** | 50.66 | 19.93 | 26.33 | 38.02 | 46.15 | **5.65** |
| **PHyCLIP** | ✓ | **44.43** | **89.30** | **59.83** | **56.18** | **75.76** | **95.06** | **56.81** | **16.00** | **10.47** | 3.05 | **54.64** | 20.41 | **26.44** | 33.43 | **50.13** | 5.42 |

The best and second performances are emphasized by bold fonts and underlines, respectively.

## Table 2: Zero-shot retrieval and hierarchical classification.

| | w/ boxes | COCO R@5 | COCO R@10 | Flickr R@5 | Flickr R@10 | COCO R@5 | COCO R@10 | Flickr R@5 | Flickr R@10 | TIE($\downarrow$) | LCA($\downarrow$) | $J(\uparrow)$ | $P_H(\uparrow)$ | $R_H(\uparrow)$ |
|---|---|---|---|---|---|---|---|---|---|---|---|---|---|---|
| | | Text $\rightarrow$ Image | | | | Image $\rightarrow$ Text | | | | Hierarchical metrics (WordNet) | | | | |
| **ViT B/16** CLIP | | 56.39 | 67.59 | 83.30 | 89.70 | **70.44** | 80.42 | **93.10** | **95.70** | 3.705 | 2.254 | 0.7805 | 0.8498 | 0.8503 |
| MERU | | 55.50 | 66.71 | 82.26 | 88.84 | 69.32 | 78.96 | 89.70 | **95.70** | 3.832 | 2.292 | 0.7720 | 0.8451 | 0.8439 |
| HyCoCLIP | ✓ | 56.24 | 67.69 | 82.90 | 88.94 | 69.00 | 79.16 | 91.90 | 95.30 | 3.378 | 2.113 | 0.8008 | 0.8653 | 0.8636 |
| **PHyCLIP** | ✓ | **58.00** | **68.74** | **83.40** | **89.92** | 70.20 | **80.44** | 91.10 | 95.60 | **3.285** | **2.088** | **0.8065** | **0.8684** | **0.8682** |

all more specific concepts $x \preceq y$ are expected to belong. We measure violations using the exterior angle $\phi(x, y)$ and penalize cases where $x \notin C(y)$ through the entailment loss :

$$L_{\text{ent},i}(X, Y) = \max(0, \phi(x^{(i)}, y^{(i)}) - \eta\omega(y^{(i)})), \quad L_{\text{ent}}(X, Y) = \frac{1}{k}\sum_{i=1}^{k} L_{\text{ent},i}(X, Y), \quad (4)$$

with the margin size $\eta$ (Pal et al., 2025). We sum this loss over known pairs and get $\mathcal{L}_{\text{ent}}$.

$$\mathcal{L}_{\text{ent}} = \sum_{b \in B} \left( L_{\text{ent}}(I_b, T_b) + L_{\text{ent}}(I_b^{\text{box}}, T_b^{\text{box}}) + L_{\text{ent}}(I_b, I_b^{\text{box}}) + L_{\text{ent}}(T_b, T_b^{\text{box}}) \right) \quad (5)$$

Finally, the overall training objective is the sum of the two weighted by $\gamma$:

$$\mathcal{L}_{\text{overall}} = \mathcal{L}_{\text{cont}} + \gamma\mathcal{L}_{\text{ent}}, \quad (6)$$

# 3 Experiments

## 3.1 Training Details

We trained on the GRIT dataset (Peng et al., 2023), which provides automatically annotated image–text pairs with bounding boxes. Although only 14.0M pairs with 26.6 million box annotations were available due to outdated links (vs. 20.5 million with 35.9 million box annotations in the original paper), the scale remains much larger than manually annotated datasets such as Flickr30K Entities (Plummer et al., 2015). We compare against CLIP (Radford et al., 2021), MERU (Desai et al., 2023), and HyCoCLIP (Pal et al., 2025), all trained on GRIT from scratch under the same protocol as HyCoCLIP. For PHyCLIP, we use $k = 64$ factors with $d = 8$ dimensions each (see Appendix B for ablation studies, and Appendix C for implementation details). While PHyCLIP incorporates multiple factors, each factor is low-dimensional, and the factor-wise distances can be computed in parallel; the computational cost of PHyCLIP is the same as HyCoCLIP.

## 3.2 Experimental Results and Analysis

**Zero-shot Image Classification.** We evaluated zero-shot image classification on 16 datasets following the protocol proposed by CLIP (Radford et al., 2021), which uses General, Fine-grained, and Specialized datasets. Table 1 summarizes the results. PHyCLIP obtained consistent performance gains, particularly on General datasets. This suggests that PHyCLIP successfully assigns concept families to hyperbolic factors and supports coarse-grained classification. Within Fine-grained datasets, improvements on Food-101 (Bossard et al., 2014) and Oxford Pets (Parkhi et al., 2012) are remarkable, suggesting that intra-family taxonomies are effectively captured without confusion across families. While not best on every dataset, the gaps on Flowers-102 (Nilsback & Zisserman, 2008) are small, FGVC-Aircraft (Maji et al., 2013) and Country211 (Radford et al., 2021) remain challenging

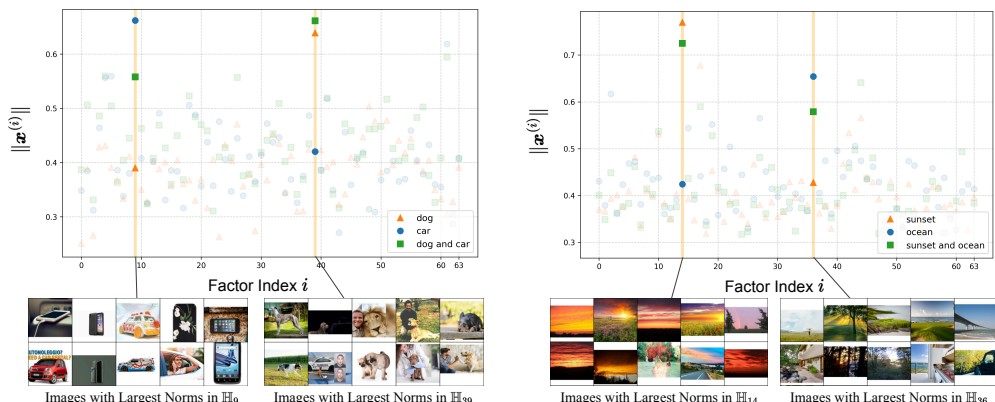

Figure 2: Factor-wise norms for single and composite concepts. PHyCLIP learns to represent the conjunction (left) "a dog and a car" and (right) "a sunset and the ocean" by simultaneously activating the specialized factors for each concept, analogous to a Boolean algebra.

due to extreme intra-class similarity, and EuroSAT (Helber et al., 2019) is out-of-distribution domain for the GRIT dataset used for pretraining. Overall, PHyCLIP achieves the strongest performance among comparison models.

**Zero-shot Image and Text Retrieval.** We evaluate cross-modal alignment by zero-shot retrieval on COCO (Lin et al., 2014) and Flickr30K (Young et al., 2014; Karpathy & Fei-Fei, 2015) and report Recall@$k$ in Table 2. PHyCLIP achieves the best performance across all metrics and datasets for image retrieval, which validates the $\ell_1$-product metric in Eq. (1) penalizes mismatches across hyperbolic factors more effectively than a single hyperbolic space. For text retrieval, CLIP remains strong and PHyCLIP shows competitive results.

**Hierarchical Classification.** We evaluate the expressivity of *is-a* taxonomy on ImageNet (Russakovsky et al., 2015), where class labels are enriched by WordNet (Miller, 1995) and errors are measured with standard hierarchical metrics (Kosmopoulos et al., 2015). The results in Table 2 show that PHyCLIP consistently attains superior scores, demonstrating not only higher top-1 accuracy but also that misclassifications remain close to the ground-truth class in the taxonomy. This suggests that our factorized design better preserves semantic neighborhoods: the $\ell_1$-product metric handles cross-family compositionality at the Boolean-lattice level, while each hyperbolic factor is free to model cleaner intra-family *is-a* hierarchies. As a result, PHyCLIP yields disentangled, hierarchy-aligned representations that validate the effectiveness of our approach.

**Composition via $\ell_1$-Product Metric.** To study how the $\ell_1$-product metric encodes compositionality, we compare embeddings of two single-concept prompts ("a photo of dog", "a photo of car") with their conjunctive prompt ("a photo of a dog and a car"). Figure 2 shows factor-wise norms over $k = 64$ factors. "Dog" peaks at factor $i = 39$ while "car" peaks at $i = 9$. Images with the largest norms in these factors demonstrate that factor $i = 39$ captures the family of mammals while $i = 9$ captures the family of daily-use artifacts. Moreover, the conjunctive prompt "a dog and a car" yields large norms in *both* factors. This indicates that a composition leads to simultaneous activation of concept-specific factors. Similar pattern is observed for "a sunset and the ocean" (see the right panel of Fig. 2). This behavior resembles Boolean algebra: composing concepts corresponds to the union of subsets, or equivalently the element-wise $\max$ of binary indicators.

## 4 Conclusion

We introduced PHyCLIP, a vision–language model that learns representations using an $\ell_1$-product metric on a Cartesian product of hyperbolic spaces. We empirically demonstrated that PHyCLIP simultaneously captures the compositionality across concept families through the $\ell_1$-product metric analogous to a Boolean algebra, as well as *is-a* taxonomies within hyperbolic spaces in the way of hyperbolic embeddings. This design yields state-of-the-art performances across various tasks.

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

## A   Theoretical Background

**Geometry and Embedding of Hierarchies.** Hierarchical relations, such as *is-a* or entailment structures, correspond to tree structures and are naturally modeled by hyperbolic geometry, which embeds trees with arbitrarily small distortion.

**Theorem 1** (Hyperbolic embedding of trees (Sarkar, 2011))**.** *Let $\mathbb{H}^d$ be a $d$-dimensional hyperbolic space with the hyperbolic distance $d_{\mathbb{H}^d}$. For every finite metric tree $T$ (and every infinite metric tree $T$ with known bounds for maximum degree and minimum edge length), and for every $\varepsilon > 0$, there exist a scale $\tau > 0$ and an embedding $f : \tau T \to \mathbb{H}^2$ such that the distortion is at most $1 + \varepsilon$; that is, there exists a $(1 + \varepsilon, 0)$-quasi-isometric embedding $f$ up to scaling.*

See Theorem 5 in Sarkar (2011) for the proof. This explains the empirical success of hyperbolic embeddings for hierarchical data (Nickel & Kiela, 2017, 2018; Ganea et al., 2018; Sala et al., 2018; Tifrea et al., 2019). In practice, $\mathbb{H}^d$ with $d > 2$ is common for achieving better performance.

**Geometry and Embedding of Compositionality.** Compositional structures, such as concept conjunctions like "a dog in a car," correspond to Boolean lattices and are naturally modeled by $\ell_1$-product metrics, which capture the additive combination of independent components.

**Definition 1** ($\ell_1$-product metric space)**.** *Let $\{(X_i, d_i)\}_{i=1}^{k}$ be non-trivial metric spaces. An $\ell_1$-product metric space of $\{(X_i, d_i)\}_{i=1}^{k}$ is a Cartesian product space $\prod_{i=1}^{k} X_i$ equipped with the $\ell_1$-product metric $(\sum_{i=1}^{k} d_i)((\boldsymbol{x}^{(1)}, \ldots, \boldsymbol{x}^{(k)}), (\boldsymbol{y}^{(1)}, \ldots, \boldsymbol{y}^{(k)})) = \sum_{i=1}^{k} d_i(\boldsymbol{x}^{(i)}, \boldsymbol{y}^{(i)})$. If not ambiguous even without subscripts, this space is denoted by $(X^k, d_{X^k})$ for brevity.*

**Proposition 1** (Embedding of Boolean Lattice)**.** *A Boolean lattice $(2^{\mathcal{C}}, \preceq)$ for $n$ atomic concepts can be embedded into the poset $(\mathbb{R}^n, \preceq)$ while preserving the order relations. As a metric space $(\{0, 1\}^n, d_{\text{Ham}})$, it is isometrically embedded into an $\ell_1$-product metric space $(\prod_{i=1}^{k} X_i, \sum_{i=1}^{k} d_i)$ for any $k \geq n$ after appropriate per-factor scaling. However, it admits no isometric embedding into a hyperbolic space $\mathbb{H}^d$ for any $d \geq 2$ and $n \geq 2$.*

263 See Proposition I.4 in Bridson & Haefliger (1999) for the proof. This confirms that $\ell_1$-product metrics,
264 not hyperbolic geometry, provide the appropriate framework for compositional structures.

## B Ablation Study

266 We investigate the contributions of embedding space
267 factorization and the $\ell_1$-product metric through ablation
268 studies, summarized in Table 3. We fix the total embed-
269 ding dimension $kd$ and vary the number of factors, $k$.
270 When $k = 1$ (equivalent to HyCoCLIP), performance is
271 the lowest on most metrics; increasing $k$ generally im-
272 proves results, except for text retrieval, thereby demon-
273 strating the benefit of factorization. Performance peaks
274 at $k = 64$ or $k = 128$, although zero-shot classifica-
275 tion accuracy for Food-101 (Bossard et al., 2014) drops
276 substantially at $k = 128$, indicating that overly fine fac-
277 torization may impair the representation of intra-family
278 taxonomy. Replacing the $\ell_1$-product metric with the

Table 3: Ablation study.

| # of factors, $k$ | # of dims., $d$ | product metric | classification | | retrieval COCO, R@5 | | hierarchical | |
|---|---|---|---|---|---|---|---|---|
| | | | ImageNet | Food-101 | Image | Text | TIE | J |
| 1 | 512 | – | 42.93 | 51.71 | 56.24 | 69.00 | 3.378 | 0.8008 |
| 8 | 64 | $\ell_1$ | 44.26 | 52.16 | 57.28 | 69.38 | 3.288 | 0.8061 |
| 16 | 32 | $\ell_1$ | 44.03 | 54.89 | 56.78 | 67.62 | 3.292 | 0.8063 |
| 32 | 16 | $\ell_1$ | 43.90 | 54.48 | 56.70 | 66.92 | 3.324 | 0.8035 |
| 64 | 8 | $\ell_1$ | 44.43 | 56.81 | 58.00 | 70.20 | 3.285 | 0.8065 |
| 128 | 4 | $\ell_1$ | 44.08 | 52.61 | 57.82 | 71.44 | 3.278 | 0.8073 |
| 64 | 8 | $\ell_2$ | 43.46 | 51.44 | 57.72 | 71.40 | 3.377 | 0.7998 |

279 Riemannian ($\ell_2$) product metric consistently degrades performance, except for text retrieval. This
280 result supports that the $\ell_1$-product metric provides a more effective way to aggregate cross-family
281 composition.

## C Implementation Details

### C.1 Model Architecture and Hyperparameters.

284 We introduce the details of our implementation and hyperparameters, which follow Desai et al. (2023);
285 Pal et al. (2025).

286 As an image encoder, we employ the base Vision Transformer (Dosovitskiy et al., 2021; Chen et al.,
287 2021; Touvron et al., 2021) with a patch size of 16. Each image is randomly resized by a scale
288 from 0.5 to 1.0 and randomly cropped to $224 \times 224$ pixels, resulting in 196 tokens, concatenated
289 with 2-D sine-cosine position embeddings. We employ the text encoder used by the original CLIP
290 (Radford et al., 2021), which consists of a 12-layer Transformer architecture (Vaswani et al., 2017)
291 with 512 dimensions. The outputs of image and text encoders are scaled by learnable scalars
292 $c_{\text{img}}$ and $c_{\text{txt}}$, respectively, before mapped by the exponential map. These scalars are initialized as
293 $c_{\text{img}} = c_{\text{txt}} = 1/\sqrt{512}$.

294 For the contrastive loss $L_{\text{cont}}$ in Eq. (2), the temperature $\tau$ is initialized as $0.07$ and clipped at a
295 minimum value of $0.01$. For the entailment loss $L_{\text{ent}}$ in Eq. (4) , the hyperparameter $\eta$ is set to $\eta = 0.7$
296 for inter-modality entailments ($I \preceq T$ and $I^{\text{box}} \preceq T^{\text{box}}$) and $\eta = 1.2$ of intra-modality entailments
297 ($T \preceq I^{\text{box}}$ and $T \preceq T^{\text{box}}$). The hyperparameter $\gamma$ for the overall loss in Eq. (6) is set to $\gamma = 0.2$.

298 We trained each model on 4 A100 GPUs for 500,000 iterations with a batch size of 768. We used
299 the AdamW optimizer (Loshchilov & Hutter, 2019) with hyperparameters $\beta_1 = 0.9, \beta_2 = 0.98$
300 and weight decay $0.2$ (only for model parameters but not for scalars such as $\tau$, $c_{\text{img}}$, $c_{\text{txt}}$, $\alpha_i$). We
301 used a cosine learning rate scheduler (Loshchilov & Hutter, 2017) with a maximum learning rate of
302 $5 \times 10^{-4}$ and a warm up for 4,000 steps.

### C.2 Benchmarks

304 **Zero-shot Image Classification.** Each class is accompanied by short text descriptions such as "a
305 photo of a {class name}". The prediction is made by selecting the class whose description is closest
306 to the image in the embedding space. Please refer to the protocol in Desai et al. (2023). We evaluate
307 on the following datasets:

308 • **ImageNet** (Russakovsky et al., 2015)

309 • **Food101** (Bossard et al., 2014)

310 • **CIFAR-10** (Krizhevsky & Hinton, 2009)

- **CIFAR-100** (Krizhevsky & Hinton, 2009)
- **CUB-2011** (Wah et al., 2011)
- **SUN397** (Xiao et al., 2010)
- **Stanford Cars** (Krause et al., 2013)
- **FGVC Aircraft** (Maji et al., 2013)
- **DTD** (Cimpoi et al., 2014)
- **Oxf-IIIT Pets** (Parkhi et al., 2012)
- **Caltech-101** (Fei-Fei et al., 2004)
- **Flowers-102** (Nilsback & Zisserman, 2008)
- **STL-10** (Coates et al., 2011)
- **EuroSAT** (Helber et al., 2019)
- **RESISC45** (Cheng et al., 2017)
- **Country211** (Radford et al., 2021)

**Zero-shot Image and Text Retrieval.** We follow the retrieval protocol of Desai et al. (2023), using:

- **COCO** (Lin et al., 2014)
- **Flickr30K** (Young et al., 2014; Karpathy & Fei-Fei, 2015)

**Hierarchical Classification.** We follow the protocol of Pal et al. (2025) based on ImageNet (Russakovsky et al., 2015) and WordNet (Miller, 1995).

