# OpenReview forum: "Product of Hyperbolic Spaces  for Vision-Language Representation Learning"
_EurIPS.cc/2025/Workshop/UPLB — Submitted to UPLB2025_

### Official Review · Reviewer_xmLV · 2025-10-30
**Review on "Product of Hyperbolic Spaces for Vision-Language Representation Learning"**

**Rating:** 3
**Confidence:** 4

**Review:**

The author outlines a method to better represent the hierarchical and compositional nature of images and text at the level of embeddings, the context of the work being encoder-decoder models such as Transformers. Their strategy is based on a decomposition of the embedding space in cartesian products of hyperbolic spaces.

The goal of the work seems interesting but the exposition is not very clear for non-experts of the topic. The text is too concise, even when additional explanations and/or discussions would be necessary. Most importantly, nowhere in the text I could find a discussion on bias in learning, data and architecture (in fact, the very root 'bias' is completely absent from the text).

Thus, since the paper does not seem to align with the topic of the workshop, it needs to be rejected.

---

### Decision · Program_Chairs · 2025-11-03

Reject